# The Role of the Global Solar UV Index for Sun Protection of Children in German Kindergartens

**DOI:** 10.3390/children9020198

**Published:** 2022-02-03

**Authors:** Olaf Gefeller, Sonja Mathes, Wolfgang Uter, Annette B. Pfahlberg

**Affiliations:** 1Department of Medical Informatics, Biometry and Epidemiology, Friedrich-Alexander-University of Erlangen-Nuremberg, 91054 Erlangen, Germany; wolfgang.uter@fau.de (W.U.); annette.pfahlberg@fau.de (A.B.P.); 2Department of Dermatology and Allergy, Technical University of Munich, 80802 Munich, Germany; sonja.mathes@mri.tum.de

**Keywords:** health attitudes, health knowledge, skin cancer, sun protection, ultraviolet radiation, UV index

## Abstract

More than twenty-five years ago, the Global Solar UV index (UVI) was introduced as a simple means of visualizing the intensity of ultraviolet radiation and to alert people to the need for sun protection. In our survey, among directors of 436 kindergartens in southern Germany we investigated the level of awareness and knowledge about the UVI as well as the practical consequences for sun protection in kindergartens. Less than half of the directors (*n* = 208, 47.7%) had ever heard of the UVI, and only a small minority of them (*n* = 34, 8.7%) used the daily UVI information to adapt sun protective measures in their kindergartens. Detailed knowledge about the UVI was a rarity among the respondents. The proportion of respondents with self-perceived detailed UVI knowledge was five times higher than the actual knowledge assessed by an in-depth structured interview using open-ended questions about the UVI (14.2% vs. 2.8%). No clear relationship between UVI awareness, knowledge, and use and directors’ age and gender was found. The UVI-related variables also showed no association with directors’ knowledge of risk factors for skin cancer and their attitudes towards tanned skin. Overall, the results paint a sobering picture regarding the penetration of the UVI into sun protection policies of German kindergartens. Future public health campaigns should aim to increase the awareness and understanding of the UVI as well as its relevance for sun protection of children.

## 1. Introduction

Exposure to intense ultraviolet (UV) radiation, especially during the early years of life, has been established as a risk factor for skin cancer [1,2,3,4,5,6,7]. Studies have shown that a substantial proportion of skin cancer cases can be attributed to UV overexposure [8]. The incidence of common types of skin cancer (cutaneous melanoma and keratinocyte carcinomas) has been rising for decades [9,10,11]. According to the most recent official cancer registry data, nearly 200,000 incident cases of keratinocyte carcinomas and nearly 23,000 incident cases of cutaneous melanoma occurred in 2018 in Germany [12]. This development was mainly caused by (i) a change in outdoor activities accompanying an increased amount of leisure time and (ii) a change in attitude towards tanned skin, which has become a symbol of attractiveness, health and fashion over recent decades [13,14]. These changes led to a higher exposure to UV radiation of a growing part of the population, mostly in the form of intermittent overexposure to the sun instead of a chronic sun exposure, which was previously dominant when outdoor occupations were the main reason for exposure [15]. Considering these developments, skin cancer prevention has nowadays become one of the most important public health issues in all countries with a predominantly fair-skinned (Caucasian) population; however, not all of these countries have recognized the dimension of this public health problem [16,17]. The primary cornerstone of skin cancer prevention is the appropriate management of sun exposure in order to limit UV exposure. The concept of public health photoprotection, which includes avoiding UV radiation from artificial sources for intentional tanning, limiting outdoor activities during the UV peak time around solar noon, seeking shade, wearing appropriate sun protective clothing (including wide-brimmed sunhats and UV-opaque sunglasses) and applying sunscreens with a high sun protection factor on uncovered skin areas, is crucial to reduce or even prevent the potential harms associated with UV overexposure [18].

As UV radiation cannot be perceived directly by humans, identifying those outdoor situations in which sun protection is necessary and finding the appropriate protection level intuitively is hardly possible. To foster sun protection at the population level a simple tool to visualize the intensity of UV radiation would be helpful. Thirty years ago, scientists at the Canadian Federal Department of the Environment (now called Environment and Climate Change Canada) created the concept of an index capturing this information in a simple integer value [19]. In 1995, the World Health Organization (WHO), together with the World Meteorological Organization (WMO), the International Commission on Non-Ionizing Radiation Protection (ICNIRP) and the United Nations Environment Programme (UNEP), adopted a slightly modified version of the Canadian index as the Global Solar UV Index (UVI) to provide uniform information to the public worldwide [20].

The UVI constitutes a measure of the daily maximum intensity of erythemally weighted solar UV irradiance [21,22] or, in plain words, the potential of the prevailing UV radiation to induce sunburn. Surveys in the initial years following the introduction of the UVI suggested that many people did not understand the UVI and hence failed to translate recommendations of sun protection based on the UVI into practice [23,24]. Subsequently, the WHO and partner organizations published a practical guide in 2002 [25] to promote the use of the UVI as an educational tool. In this practical guide, they suggested a harmonized UVI reporting scheme and attached specific protective recommendations to different UVI levels. The WHO categories of the UVI and their corresponding sun protection messages are: (i) “low” UVI levels (≤2), with no specific protective recommendation, (ii) “moderate” (3–5) and “high” (6–7) UVI values, with the recommendation to apply all sun protection measures including staying in the shade (during midday hours), clothing, sunscreen, sunglasses and a wide-brimmed hat, (iii) “very high” (8–10) and “extreme” (11+) UVI levels, where people should use all of the aforementioned sun protection measures, seek shade all day and avoid being outdoors during peak UV hours around solar noon.

In Germany, the responsibility for sun protection during early childhood is often shared between parents and caregivers at kindergartens [26]. While the parental role has been examined intensively in numerous qualitative and quantitative studies (see the two recent publications [27,28] for references), the caregiver’s perspective has less often been scrutinized. The specific aspect of the impact of the UVI on sun protection measures in German kindergartens has, to the best of our knowledge, never been addressed before. Only if caregivers are aware of the UVI and the information and preventive prompts provided by it can they adequately adapt their sun protective guidance for the children under their supervision. As the implementation of sun protection policies is the responsibility of the director of the kindergarten, we focused on this target group in our survey. We assessed the awareness of, and knowledge about, the UVI as well as the consequences of UVI information for daily sun protection in German kindergartens by structured personal interviews with the directors of kindergartens.

## 2. Materials and Methods

### 2.1. Study Setting and Participants

The study region was located in the northern part of Bavaria, a federal state in southern Germany, and comprised eight contiguous (urban and rural) districts with a population of 1.21 million. The largest city in the study region was Nuremberg with roughly 518,000 inhabitants. Using the official administrative data of the municipalities, we were able to identify 641 kindergartens in the study region. Six forest kindergartens (i.e., kindergartens where children are cared for almost all the time outdoors in a forest or another natural environment) and three kindergartens that had already been visited in the pilot phase of the study to check feasibility of the study procedures, applicability of the guideline for the structured interviews, and comprehensibility of the questions with directors and staff members of these kindergartens were excluded. We approached all directors of the remaining 632 kindergartens by postal mail. In the letter we informed them about the study and announced a later phone call. We contacted the kindergartens by phone to answer questions and to request participation. After consent to participate was given, an appointment was made for a personal interview with the kindergarten management. The study was approved by the Ethics Committee of the University of Erlangen-Nuremberg (14-059-B, 25 March 2014).

### 2.2. Assessment of Information

During the visits to the kindergartens in the spring (starting mid-April) and summer months up to September 2016, personal structured interviews were conducted with the directors by trained interviewers. As part of the interviews awareness, knowledge, and use of the UVI were assessed using two protocols. General awareness of the UVI was captured identically across the total study group by the question “Have you ever heard of the UV index?”, which was asked in all interviews. The impact of the UVI information on sun protection, i.e., the practical consequences for the application of sun protective measures in the kindergartens based on the daily UVI value, was also recorded identically in all interviews by the question “Does the actual UVI value influence sun protection at your kindergarten?”, which was only asked if awareness of the UVI has been confirmed before. However, the correctness of UVI-related knowledge was determined differently. In one subgroup (*n* = 246), the correctness of knowledge about the practical implications of different UVI levels was assessed in detail using open-ended questions in a structured interview. Those who had heard the term “UV index” before were asked to explain the meaning of the term in more detail. An answer was classified as correct if it linked the UVI to the intensity of UV radiation, reflected knowledge of the range of UVI values, and attached specific preventive recommendations to the respective UVI levels. For the latter, slight discrepancies between the WHO recommendations and responses were tolerated, e.g., if a director answered that the UVI is a measure of the strength of UV radiation with a typical range of values between 0 and 10 in Germany and claimed that for a UVI of 9 and above, increased sun protection measures with avoidance of spending time outdoors around noon are necessary, then this answer was classified as correct, although the WHO recommends starting increased protection at a UVI of 8. In the remaining subgroup of interviews (*n* = 190), the data on UVI knowledge only reflect the self-assessment by the interviewed directors. In this subgroup, directors had the choice between the following three options to answer the question “What does the term “UV index” mean to you?”: (i) I have never heard the term before, (ii) I have heard the term before but cannot explain its meaning, (iii) I know the term and can explain its meaning. In the latter subgroup only, we also asked about the source of information for daily UVI values used by the directors to distinguish between occasional, passive and active information seeking. 

We also assessed the directors’ level of knowledge about risk factors for skin cancer following a procedure that has been used in self-administered questionnaires in previous studies [29,30,31]. To this end, we asked the directors to judge whether or not nine specific exposures had an impact on skin cancer risk. The nine exposures comprised six true risk factors (constitutional factors and behavioral factors related to UV radiation exposure), but also included three exposures (e.g., allergies) that have attracted public health concern, but have not been identified as skin cancer risk factors. These distracting factors served to indicate whether directors had specific knowledge about skin cancer. Details on the nine exposures queried in the interview and directors’ responses to each single exposure can be found in the Appendix A. The pattern of answers was then summarized by a sum score measuring the ability to discriminate true risk factors from other factors of public health concern unrelated to skin cancer. In more detail, the identification of a true risk factor increased the score by one point, whereas misreporting a distracting factor as risk factor decreased the score by two points. We used this unequal weighting of correct and incorrect judgements as the number of distracting factors was only half the number of true risk factors among the nine items. The resulting score had better psychometric properties, i.e., Cronbach’s alpha was higher for this definition than for other score definitions, e.g., the one using an equal-weighting scheme; see [31] for more details. The score was subsequently classified into three categories (“low” (≤1), “medium” (2,3), and “high”(4–6)).

Directors’ opinions on tanned skin were ascertained using two questions. The first question addressed the association between tanned skin and beauty by asking the directors whether they agree with the statement “When I am tanned, I feel more beautiful”. Answers were recorded on a four-point Likert scale (agree completely, agree partially, disagree partially, and disagree completely). The second question examined the association between tanned skin and health by asking the directors whether they agree with the statement “Tanned skin is healthy skin”. Again, answers were recorded on the same four-point Likert scale as for the first question.

### 2.3. Statistical Analysis

Descriptive and analytical statistical methods were used to report the data of the survey. For categorical data, absolute frequencies and proportions were computed. To visualize the precision of proportions we added 95% confidence intervals (CI) calculated using Wilson’s method which is superior to the standard Wald-CI in terms of coverage probability for small samples [32]. For the continuous variable directors’ age, the mean and standard deviation were computed to describe the age distribution. Age values were then categorized into tertiles (<42, 42–52, >52) for further analysis. Data from the two 4-point Likert scales were dichotomized (agree completely and agree partially vs. disagree partially and disagree completely) for further analysis. Statistical comparisons between groups were performed using the Chi-Quadrat test and—in cases of small numbers in the contingency tables—Fisher’s exact test. *p*-values smaller than 0.05 were interpreted as indicating statistical significance. In view of the explanatory nature of the analyses of this survey, we refrained from alpha-adjustment techniques to account for multiple statistical tests. All computations were performed utilizing SAS Version 9.4 (SAS Institute Inc., Cary, NC, USA). 

## 3. Results

### 3.1. Characteristics of Participating Kindergartens and Directors

Of the 632 contacted kindergartens, 437 (=69%) agreed to participate in the study. One kindergarten could not be visited due to logistical reasons. The remaining 436 kindergartens comprised a total of 3424 staff members and cared for 24,975 children, most of them aged 2.5 to 6 years. One hundred forty-two (33%) of the kindergartens were small (<50 attending children), 257 (59%) were medium sized (50–99 children) and 37 (8%) were large (≥100 children). 

Regarding the type of sponsorship affecting the administrative and financial background of the kindergartens in our survey, the majority were run by churches or confessional welfare associations (*n* = 231, 53%), less than a quarter were operated by the respective city or municipality (*n* = 95, 22%), national non-confessional welfare associations governed 43 (10%) kindergartens, a variety of regional registered non-profit associations owned 25 (6%) kindergartens, private non-profit limited liability companies operated 22 (5%) facilities, 15 (3%) were run by parents’ initiatives, while for the remaining five kindergartens private individuals (*n* = 3, 0.7%) and private companies (*n* = 2, 0.5%) were responsible.

The interviewed directors were predominantly female (*n* = 417, 96%), only 19 (4%) male directors were part of the study sample. The average age of the directors was 45.9 years old (interquartile range: 24.6, 54.5). 

### 3.2. Awareness of the UVI 

Nearly half of the directors of kindergartens (*n* = 208, 47.7%, 95%-CI: 43.1–52.4) stated that they have ever heard the term UV index. Table 1 gives further details of the proportions of the directors being aware of UVI in subgroups depending on sex and age. 

### 3.3. Use of the UVI 

Only 38 directors confirmed that the actual UVI influences sun protective measures at their kindergartens meaning that in less than one tenth of the kindergartens the UVI played a role in decisions regarding sun protection. Some non-significant increase in the proportion of kindergartens using the UVI information in their decision was seen with increasing age of directors (see Table 2).

### 3.4. Knowledge about the UVI 

Remarkable differences in the level of UVI knowledge between self-assessment and interviewer assessment were observed (see Table 3). While 14% of the directors self-assessed their knowledge about the UVI as detailed enough to be able to explain the concept, only 3% of directors actually presented detailed and correct UVI knowledge during the interviews. Because the two modes of determining the level and correctness of UVI knowledge in the study were used in disjunctive subgroups, we could not directly quantify directors’ overconfidence in their UVI knowledge. Notwithstanding, the difference in the proportions between the two groups of directors, i.e., those who assume they have correct UVI knowledge in the self-assessment and those who demonstrate correct UVI knowledge during the interviews, is statistically significant (14.2% vs. 2.8%, *p* < 0.0001). A substantially higher proportion of male directors than female directors had correct UVI knowledge, but due to the small number of male directors in the study, the difference achieved statistical significance only in the analysis of interviewer-assessed UVI knowledge (25.0% (male) vs. 2.1% (female), *p* = 0.02).

### 3.5. Information Sources about the UVI

The source of information on daily UVI values was determined only in a subgroup of directors (*n* = 190). The majority of directors (59%) who were aware of the UVI replied that they did not have regular information on the daily UVI; this proportion was, of course, much lower (28%) when the total group of directors was considered. Weather forecasts on TV or radio channels were named by only nine directors (=10% among the directors being aware of the UVI and 5% in the total group, respectively). The dominant source of information on daily UVI values was the internet: 29 (=32% among the directors being aware of the UVI and 15% in the total group, respectively) obtained their information actively from weather apps on the smartphone or websites with detailed weather information.

### 3.6. Relationship between Knowledge about Risk Factors for Skin Cancer and UVI-Related Variables 

None of the UVI-related variables showed a significant association with the knowledge score summarizing the directors’ knowledge about risk factors of skin cancer. Directors being uninformed about skin cancer risk factors were aware of the UVI and used it for sun protection measures at their kindergartens in similar numbers as better informed directors. The correct understanding of the UVI—self-assessed and interviewer-assessed—was even more common in the former subgroup, but not significantly different from the other subgroup (see Table 4). 

### 3.7. Relationship between Attitudes towards Tanned Skin and UVI-Related Variables 

Directors’ attitudes towards tanned skin were mixed. Two-thirds of them agreed, partially or completely, with the statement that tanned skin is beautiful, while one-sixth of them disagreed partially and completely, respectively. The statement that tanned skin is healthy was agreed with completely by only 1% of the directors, but partial agreement was stated by another 19%, the majority of directors disagreed partially (38%) or completely (42%) with this claim. No consistent picture emerged from the analysis of the relationship between the UVI- related variables, i.e., awareness, use, and knowledge of the UVI, and the responses given to the two statements regarding tanned skin (see Table 5). The awareness of the UVI was very similar in the subgroups of directors agreeing and disagreeing with claims that tanned skin is beautiful or healthy. The use of the UVI for the decision about sun protection measures was more common among directors disagreeing with the statements that tanned skin is healthy and beautiful, respectively; however, the differences in UVI use did not reach statistical significance (*p* = 0.06 and *p* = 0.11, respectively). The relationship between UVI knowledge and attitudes towards tanned skin depended on whether the level of UVI knowledge was self-assessed or determined by interviewers. For self-assessed UVI knowledge, we found remarkable differences between the subgroups agreeing and disagreeing with statements claiming that tanned skin is healthy and beautiful, respectively; however, these differences did not reach statistical significance (*p* = 0.08 and *p* = 0.06, respectively). When the correctness of UVI knowledge was determined by interviewers, these differences did not persist.

## 4. Discussion

The UVI was introduced internationally more than 25 years ago to raise awareness and encourage the public to protect their skin from harmful doses of UV radiation, ultimately causing skin cancer. Our findings show that the degree of penetration of the UVI in German kindergartens is still unsatisfactorily low. Less than half of the directors of kindergartens in our study had ever heard of the UVI, and only a small minority used the daily UVI information to adapt sun protective measures in their kindergartens. Detailed knowledge about the UVI was an absolute rarity. Interestingly, UVI knowledge appeared to be much better when directors self-assessed their level of knowledge than when interviewers assessed the correctness of UVI knowledge based on directors’ replies to open-ended questions.

Systematic reviews on awareness, understanding, use, and impact of the UVI [23,33,34] have already found that the UVI has not yet found its way into the broader population. The UVI was intended to promote public awareness about the dangers of excessive exposure to UV radiation and to improve sun protective behavior. Research findings from more than 30 studies worldwide have, however, failed to demonstrate that the UVI has yet achieved its goals. Low levels of UVI awareness have been found in Europe, while studies conducted in Australia and New Zealand showed more promising results [23]. The understanding and more detailed knowledge of the UVI were consistently lower than awareness across all regions [23]. Positive effects of its implementation on sun protective behavior have rarely been found [33]. The report of the most recent Global Solar Index Workshop stressed that the currently available research has not yet demonstrated that the UVI is an effective method of behavior change on its own [35]. Simply increasing UVI awareness and knowledge will, however, not automatically translate into achievements regarding better sun protection, as many examples from other areas of prevention research have shown [36]. Individual and institutional changes of specific elements of health-related behavior and their framework conditions, such as sun protection in kindergartens, are a complex process in which correct health knowledge is only a necessary but not a sufficient factor.

Previous research on UVI-related aspects in the specific setting of kindergarten caregivers hosting young children is limited to a single study from Queensland in which 1383 directors and senior teachers of early childhood services participated in a mailed survey in 2002 [37]. While the overall UVI awareness in this group was very high even at that time (93% of the respondents had heard of the term UVI before), only 20% could explain correctly the meaning of a UVI value of 13. Interestingly, the self-assessed level of UVI knowledge was much higher in this study too: 79% of the respondents stated that they understand fully what the UVI is. These results from an early Australian study confirm our finding that there is a huge discrepancy between self-assessed and actual UVI knowledge, although the level of UVI knowledge is quite different between Australia and Germany. Another recent study in this setting from Portugal reported detailed data on knowledge and attitudes of caregivers of children regarding sun exposure and sun protection but omitted unfortunately UVI-related variables [38]. A further recent study from Germany provided results from a randomized trial addressing the effects of a sun protection intervention consisting of an educational workshop for kindergarten caregivers, but did not specifically report on UVI-related aspects [39].

Several studies in Germany have investigated aspects of sun protection in early childhood but focused on the parents of children being supervised in kindergartens [40,41,42,43,44,45,46,47,48,49]. In the cross-sectional study by Klosterman et al. [43], 4579 parents of children who underwent a school enrollment examination in five Bavarian regions gave information on UVI-related aspects in a self-administered questionnaire. UVI awareness was 74% in this study group, higher than in any other German study, but only 8% of the parents reported that the UVI influenced their decisions regarding their children’s sun protection, which is well in line with our data.

The reasons for the current situation of limited UVI awareness, disastrous UVI knowledge, and rare UVI use are not well understood, as studies trying to elucidate the reasons are lacking. There is an urgent need for such studies of mechanisms and models from psychological theory such as self-efficacy and barriers to behavior change as well as studies analyzing promising effective means of communicating the UVI via text messages and social media. Raising UVI awareness and understanding can be achieved by tailored health promotion campaigns that educate the public efficiently. Australia can serve as a role model for Germany and other European countries as it has demonstrated that such campaigns can be implemented successfully [50,51]. However, even if awareness and knowledge of the UVI were raised to an acceptable level, there is still much to do to promote the use of the UVI as an element of sun protection. One of the barriers to using the UVI is that people make decisions about their skin protection habitually based on temperature rather than UVI as they erroneously believe that temperature is strongly correlated with the UVI [52]. When educating the public about sun protective measures, it has to be emphasized that temperature is an insufficient proxy of the UVI and should no longer be used in decision making about sun protection. Our study yielded only anecdotal evidence of what specific misunderstandings about the UVI existed among the directors, as their responses to the open-ended question to explain the UVI were not audio taped, nor were details of the incorrect responses recorded by the interviewers. However, the anecdotal evidence was consistent with the findings of a recent qualitative study among pharmacists in Germany [53] and previous studies in Australia [52,54] that the UVI is confused with the sun protection factor (SPF) of sunscreens as well as the concept of burn times, and that detailed knowledge about the UVI scale and the preventive recommendations associated with the different UVI levels is very poor.

In the setting of kindergartens, improved sun protection is of special importance, as young children are more vulnerable for several reasons and thus need particular guidance on how to deal with exposure to UV radiation. Over the first years of life, children’s skin barrier protection remains immature and early UV exposure induces changes in the skin [55]. High UV exposure during early life has also been shown to be associated with higher nevus density, which is a strong independent marker of melanoma risk [56,57,58]. In addition, the amount of solar UV radiation received over the lifetime is not uniformly distributed over all ages. Children experience substantially higher doses of UV radiation through outdoor activities and recreation than adults, as they spent a smaller proportion of the day indoors [59]. At the same time, children have only a very limited knowledge of cause-and-effect relations regarding skin cancer [60]. They are thus unable to protect themselves adequately. Therefore, caregivers at kindergartens have a great responsibility to ensure that the children they supervise receive adequate sun protection. One element of adequate sun protective guidance is the incorporation of the UVI information into decisions about the necessary level of protection. The prerequisite for a future integration of the UVI into sun protection policies in kindergartens as an element to be actually used in the daily routine is a better understanding of the UVI and its preventive prompts for sun protection among the caregivers. This requires the better education of kindergarten staff about the concept, including practical training demonstrating how the UVI information can be implemented in the daily workflow at kindergartens.

### Limitations

The limitations of our study are rooted in its moderate sample size. Although a total sample size of 436 is not particularly small, the extremely unbalanced gender distribution of study participants resulted in a very small number of male directors in our study, which limits the statistical power to detect gender differences in UVI-related variables. In addition, the two different modes of assessing UVI knowledge were conducted in two disjunctive subgroups of the study which has roughly halved the available sample size for the analysis of aspects of self-assessed and interviewer-assessed UVI knowledge, respectively. In addition, when interpreting the results it should be kept in mind that the current situation might differ from the study results as the field phase of the study has taken place some years ago. Another limitation stems from the involvement of thirteen different interviewers during the field phase of the study. Although training sessions with all interviewers were organized to standardize study procedures, including conducting personal interviews with directors of the participating kindergartens, some form of interviewer bias cannot be ruled out. The participation of the kindergartens in the study region was acceptable; however, a response rate of 69% is far below a full census. Thus, the generalization of our findings to all kindergartens of the study region is only valid under the assumption of the absence of selection among kindergartens in the study region.

## 5. Conclusions

In conclusion, the results from our survey among directors of kindergartens provide a sobering picture regarding the penetration of the UVI in German kindergartens. Future public health campaigns should target increasing awareness and understanding of the UVI and its importance for the protection of children from the sun.

## Figures and Tables

**Table 1 children-09-00198-t001:** Awareness of the UVI among directors of kindergartens (N = 436). Absolute frequencies (N = sample size; n = number of affirmative answers), proportions (%) accompanied by confidence intervals (95%-CI) and *p*-values (*p*) comparing proportions between subgroups.

Aware of UVI					
	N	n	%	95%-CI ^1^	*p* ^2^
All	436	208	47.7	43.1–52.4	
Sex					0.66
female	417	198	47.5	42.7–52.3	
male	19	10	52.6	31.7–72.7	
Age					0.74
<42	148	74	50.0	42.5–58.0	
42–52	145	69	47.6	39.6–55.7	
>52	143	65	45.5	37.5–53.6	

^1^ Wilson CI; ^2^ Chi-Square test.

**Table 2 children-09-00198-t002:** Use of UVI information for sun protection measures at German kindergartens based on answers of directors of kindergartens (N = 436). Absolute frequencies (N = sample size; n = number of affirmative answers), proportions (%) accompanied by confidence intervals (95%-CI) and *p*-values (*p*) comparing proportions between subgroups.

Use of UVI Information								
	N	n	% ^1^	95%-CI ^2^	*p* ^3^	% ^4^	95%-CI ^2^	*p* ^3^
All	436	38	8.7	6.4–11.7		18.3	13.6–24.1	
Sex					0.99			0.69
female	417	37	8.9	6.1–12.0		18.7	13.8–24.7	
male	19	1	5.3	0.9–24.6		10.0	1.8–40.4	
Age					0.36			0.22
<42	148	9	6.1	3.2–11.2		12.2	6.5–21.5	
42–52	145	14	9.7	5.8–15.6		20.3	12.5–31.2	
>52	143	15	10.5	6.4–16.6		23.1	14.5–34.6	

^1^ Proportion calculated using the complete group in the denominator; ^2^ Wilson CI; ^3^ Chi-Square test/Fisher’s exact test; ^4^ proportion calculated using only the subgroup being aware of the UVI in the denominator.

**Table 3 children-09-00198-t003:** Self-assessed and interviewer-assessed level of knowledge about the UVI among directors of kindergartens (N = 436). Absolute frequencies (N = sample size; n = number with correct knowledge), proportions (%) accompanied by confidence intervals (95%-CI) and *p*-values (*p*) comparing proportions between subgroups.

Correct UVI Knowledge								
	N	n	% ^1^	95%-CI ^2^	*p* ^3^	% ^4^	95%-CI ^2^	*p* ^3^
Self- assessment								
All	190	27	14.2	10.0–19.9		29.3	21.0–39.3	
Sex					0.66			0.21
female	179	25	14.0	9.6–19.8		28.1	19.8–38.2	
male	11	2	18.2	5.1–47.7		66.7	20.8–93.9	
Age					0.26			0.25
<42	73	11	15.1	8.6–25.0		27.5	16.1–42.8	
42–52	59	5	8.5	3.7–18.4		20.0	8.9–39.1	
>52	58	11	19.0	10.9–30.9		40.7	24.1–59.3	
Interviewer assessment								
All	246	7	2.8	1.3–5.8		6.0	2.9–11.8	
Sex					0.02			0.06
female	238	5	2.1	0.9–4.8		4.6	2.0–10.3	
male	8	2	25.0	7.1–59.1		28.6	8.2–64.1	
Age					0.99			0.99
<42	75	2	2.7	0.7–9.2		5.9	1.6–19.1	
42–52	86	3	3.5	1.2–9.8		6.8	2.3–18.2	
>52	85	2	2.4	0.6–9.8		5.3	1.5–17.3	

^1^ Proportion calculated using the complete group in the denominator; ^2^ Wilson CI; ^3^ Chi-Square test/Fisher’s exact test; ^4^ proportion calculated using only the subgroup being aware of the UVI in the denominator.

**Table 4 children-09-00198-t004:** The relationship between awareness, use and knowledge of the UVI and the level of knowledge about risk factors for skin cancer based on answers of directors of kindergartens (N = 436). Absolute frequencies (N), proportions (%) and *p*-values (*p*) assessing the association between UVI-related variables and knowledge of risk factors for skin cancer.

	Knowledge about Risk Factors of Skin Cancer						
	Low (N = 61)		Medium (N = 216)		High (N = 159)		
	N	% ^1^	N	% ^1^	N	% ^1^	*p* ^2^
Awareness of UVI	30	49.2	106	49.1	72	45.3	0.74
Use of UVI	5	8.2	21	9.7	12	7.6	0.75
Correct self-assessed UVI knowledge	6	35.3	15	31.3	6	35.3	0.93
Correct interviewer-assessed UVI knowledge	2	15.4	4	6.9	1	2.2	0.14

^1^ Awareness and use of UVI proportions were calculated using the complete knowledge group as the denominator. For UVI knowledge only the subgroup being aware of the UVI was used; ^2^ Chi-Square test/Fisher’s exact test modified for 2 × 3 tables.

**Table 5 children-09-00198-t005:** The relationship between awareness, use and knowledge of the UVI and attitudes towards tanned skin based on answers of directors of kindergartens (N = 436). Absolute frequencies (N), proportions (%) and *p*-values (*p*) assessing the association between UVI-related variables and attitude variables.

	Tanned Skin is Beautiful					Tanned Skin is Healthy				
	Agree		Disagree			Agree		Disagree		
	(N = 292)		(N = 144)			(N = 86)		(N = 349)		
	N	% ^1^	N	% ^1^	*p* ^2^	N	% ^1^	N	% ^1^	*p* ^2^
Awareness of UVI	140	47.9	68	47.2	0.89	35	40.7	173	49.6	0.14
Use of UVI	21	7.2	17	11.8	0.11	3	3.5	35	10.0	0.06
Correct self-assessed UVI knowledge	12	21.4	15	41.7	0.06	6	54.5	21	25.9	0.08
Correct interviewer-assessed UVI knowledge	5	6.0	2	6.3	0.99	3	12.5	4	4.3	0.15

^1^ Awareness and use of UVI proportions were calculated using the complete knowledge group in the denominator, for UVI knowledge only the subgroup being aware of the UVI was used; ^2^ Chi-Square test/Fisher’s exact test.

## Data Availability

The data presented in this study are available on request from the corresponding author.

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
