# Peer review of "The Role of the Global Solar UV Index for Sun Protection of Children in German Kindergartens"

_children, 2022, doi:10.3390/children9020198_

Round 1
Reviewer 1 Report
This paper provides useful information about the knowledge of UVI in german kindergartens. The statistical analysis performed in the study and the assessment of information are adequate.
- Page 1 line 34- the authors mention the incidence of common skin cancer types- here please use plural form of "keratinocyte carcinoma"
- Page 2 line 75- "in Germany" is duplicated in the same sentence, please correct
- This study suggests the importance of implication of UVI into sun protection policies of kindergartens. How would the authors reach this goal, what would be their recommendation e.g. participating mandatory on-line training regarding UVI and sun-protection etc.
- What could be the reason that male directors had a better knowledge about UVI?
Author Response
We thank the reviewer for his/her thoughtful criticism and constructive comments on our manuscript. We have prepared a revised and expanded version of the manuscript incorporating the suggestions. We address below the specific points raised by the reviewers and explain how they have been dealt with in the revised manuscript. The highlighted version of the revised manuscript accompanying the submission shows the changes in detail.
Reviewer 1:
This paper provides useful information about the knowledge of UVI in German kindergartens. The statistical analysis performed in the study and the assessment of information are adequate.
Authors’ answer:
We thank the reviewer for the positive evaluation of the merits of our manuscript.
Page 1 line 34- the authors mention the incidence of common skin cancer types- here please use plural form of "keratinocyte carcinoma"
Authors’ answer:
We have changed to “keratinocyte carcinomas”.
Page 2 line 75- "in Germany" is duplicated in the same sentence, please correct
Authors’ answer:
We eliminated the second occurrence of “in Germany”.
This study suggests the importance of implication of UVI into sun protection policies of kindergartens. How would the authors reach this goal, what would be their recommendation e.g. participating mandatory on-line training regarding UVI and sun-protection etc.
Authors’ answer:
The prerequisite for a future integration of the UVI into sun protection policies in kindergartens is a better understanding of the UVI and its preventive guidance for daily sun protection in kindergartens. This requires better education of kindergarten staff about the concept. As a next step, the sun protection policies should be updated to include the UVI as an element to be actually used in the daily routine of sun-protective guidance. We are aware that it will take some time to implement such changes in all kindergartens, but it is time to start this process. We have expanded last paragraph (prior to the Limitations) of the Discussion section to reflect these ideas.
What could be the reason that male directors had a better knowledge about UVI?
Authors’ answer:
We can only speculate about the reasons for sex-specific differences in UVI knowledge and have refrained from including these speculations in the manuscript as a small sample artefact (due to the limited number of male directors in our study) may also be an explanation for the finding.
Reviewer 2 Report
This article is very insightful and I believe this article to be a great addition to the journal. Here are some of my considerations:
1. Page 4 row 161. Please explain the reason for specific age categorization.
2. Page 4 row 188. Please put in the number of females who were part of the study sample
Otherwise a very insightful article
Author Response
We thank the reviewer for his/her thoughtful criticism and constructive comments on our manuscript. We have prepared a revised and expanded version of the manuscript incorporating the suggestions. We address below the specific points raised by the reviewers and explain how they have been dealt with in the revised manuscript. The highlighted version of the revised manuscript accompanying the submission shows the changes in detail.
Reviewer 2:
This article is very insightful and I believe this article to be a great addition to the journal.
Authors’ answer:
We thank the reviewer for the positive evaluation of the merits of our manuscript.
Here are some of my considerations:
- Page 4 row 161. Please explain the reason for specific age categorization.
Authors’ answer:
We categorized the age variable in a data-adaptive manner into three categories, each comprising one third of the sample each. That is meant by our expression in line ??? “values were then categorized into tertiles”. Such data-adaptive categorizations have statistical advantages with respect to an increased statistical power when testing for differences between age subgroups.
- Page 4 row 188. Please put in the number of females who were part of the study sample
Otherwise a very insightful article.
Authors’ answer:
We added the absolute number of females as requested.
Reviewer 3 Report
This manuscript is very well written and explores a highly important topic among an essential target group. The methods and results section are written very well and are well specified.
I however recommend major revision, mostly due to the fact that some essential parts in the introduction and discussion are missing.
Review of manuscript titled; "The Role of the Global Solar UV Index for Sun Protection of 2 Children in German Kindergartens"
In general
The topic of awareness about the UV index and translation into actions, especially among a highly vulnerable group of young children, is very relevant. The research performed in this manuscript is important in the light of skin cancer prevention, focusing on increasing knowledge about awareness of the UVI and attitude concerning tanned skin of daycare directors. The manuscript is very well written and of high quality and provides directions for improving knowledge among kindergarten directors concerning the UVI. Although the results are interesting, it is a bit unfortunate that the results stem from five years ago. Current knowledge levels about the UVI could have changed for the better due to increased media attention in the last years. Also, some parts of interesting findings are absent in the discussion section. The undermentioned comments should be considered by the authors.
Abstract
‘…should target increasing…’ (line 25, p. 1)
- I believe this should be ‘target on’ or ‘focus on’
Introduction
In general: the introduction is written in high quality English, although some sentences are unnecessary complicated. I am missing some references to substantiate information. Moreover, some facts and numbers about skin cancer in countries and Germany specifically would be a good addition. Some contextual information about kindergartens and the choice of focusing on the target group of managers would be interesting. E.g. I would like to read something about behaviours that are being transferred/modelled to children.
‘…has mainly been triggered’ (line 35, p.1)
- This sounds a bit strange. ‘the main cause of skin cancer increase is …’ or something like that is more suitable.
- The reference which is referred to in lines 35-39 is rather old (1995) and the statement is not complete. Not all skin cancers are caused by intermittent exposure. Childhood sun exposure and sunburn are more strongly correlated with melanoma, intermittent exposure is strongly connected with melanoma and BCC, while cumulative exposure is more strongly related to SCC. There are more novel references to point out.
- Line 40: it sounds as if skin cancer prevention has already become one of the most important health problem issues in European countries, while this should become one of the most important health related issues since many countries do not have established prevention programmes. Further, references are missing.
- Line 51: intuitively or impossible? We can not see nor feel the proportion of UVI.
- It would be worthwhile to read more about the reason why tanned skin as a subject was an important part of the questionnaire.
Methods
In general: the method section is well-written, coherently and in-depth. It is very good that the investigators were able to include many kindergartens.
- What are ‘trained interviewers’? How were they trained? Were they experienced? Were they part of the investigator’s team?
Results
The study provides important insight into knowledge levels about UVI and attitudes about tanned skin that should be addressed in the future.
- It is not clear why the information concerning administrative and financial background of the kindergartens is mentioned, unless the authors would elaborate on it or interpret it further in the manuscript
- ‘… female (96%), only 19 (=4%) male’ (line 88): a typo.
- I would be interested in the percentage of correct answers about specific skin cancer risk factors; which ones stood out?
Discussion
In general: the discussion focuses on important findings. The discussion bears some unnecessary repetitive sentences, especially in the second and third paragraph – some parts can therefore be adjusted and/or shortened.
- ‘… The report of the most’ (line 321): this statement is rather logic; improving knowledge does rarely lead to behavior change. The authors could explore this a bit more.
- Since the difference between self-assessed and actual UVI knowledge is so important, I would like to read more about which misinterpretations or misconceptions were observed among the directors. With that information, a more tailored approach in interventions could be accomplished.
- ‘disastrous knowledge’ (line 348): again, if the knowledge gaps were so profound and the authors advice a tailored approach, I want to know more about the actual misconceptions.
- Greater parts of the last paragraph of the discussion (before limitations) would actually suit better in the introduction section – where the importance of protection of children at daycare is now quite restricted. Moreover, this part does not elaborate on findings in particular but is more an introductory part for selecting daycare staff as target group.
- I am missing reflection on a highly important finding; the high percentage of directors that saw tanned skin as beautiful (n=292) and as healthy (n=86). The attitude about tanned skin is now absent in the discussion section, while it is 1) a very important finding and can function as a barrier for actual sun protection behavior, and 2) a topic which has gained a lot of attention in earlier skin cancer prevention research. The authors should implement a section which discusses this imperative topic.
Author Response
Point-by-point reply to the reviewers’ comments
We thank the reviewer for his/her thoughtful criticism and constructive comments on our manuscript. We have prepared a revised and expanded version of the manuscript incorporating the suggestions. We address below the specific points raised by the reviewer and explain how they have been dealt with in the revised manuscript. The highlighted version of the revised manuscript accompanying the submission shows the changes in detail. When we refer to specific page and line numbers in our reply the revised version without the highlighted changes is the basis for these text references.
Reviewer 3:
This manuscript is very well written and explores a highly important topic among an essential target group. The methods and results section are written very well and are well specified.
I however recommend major revision, mostly due to the fact that some essential parts in the introduction and discussion are missing.
The topic of awareness about the UV index and translation into actions, especially among a highly vulnerable group of young children, is very relevant. The research performed in this manuscript is important in the light of skin cancer prevention, focusing on increasing knowledge about awareness of the UVI and attitude concerning tanned skin of daycare directors. The manuscript is very well written and of high quality and provides directions for improving knowledge among kindergarten directors concerning the UVI.
Authors’ answer:
We thank the reviewer for the – at least overall - positive evaluation of the merits of our manuscript.
Although the results are interesting, it is a bit unfortunate that the results stem from five years ago. Current knowledge levels about the UVI could have changed for the better due to increased media attention in the last years. Also, some parts of interesting findings are absent in the discussion section. The undermentioned comments should be considered by the authors.
Authors’ answer:
We have added a statement to the paragraph devoted to limitations dealing with the possible outdated nature of our results (see line 416-418).
Abstract
‘…should target increasing…’ (line 25, p. 1)
- I believe this should be ‘target on’ or ‘focus on’
Authors’ answer:
The sentence has been rephrased. It now reads, “Future public health campaigns should aim to increase awareness and understanding of the UVI as well as its relevance for sun protection of children.”
Introduction
In general: the introduction is written in high quality English, although some sentences are unnecessary complicated. I am missing some references to substantiate information. Moreover, some facts and numbers about skin cancer in countries and Germany specifically would be a good addition. Some contextual information about kindergartens and the choice of focusing on the target group of managers would be interesting. E.g. I would like to read something about behaviours that are being transferred/modelled to children.
Authors’ answer:
We thank the reviewer for his/her comment. We have added seven references to substantiate claims made in the Introduction. As requested we also incorporated some brief epidemiologic information on skin cancer in Germany (see line 34-36). We expanded the final paragraph of the Introduction slightly to motivate the choice of the setting kindergarten and the choice of the target group (directors). To our opinion, the Introduction should not be too long. Therefore, we tried to be brief and left intentionally some issues for the Discussion.
‘…has mainly been triggered’ (line 35, p.1)
- This sounds a bit strange. ‘the main cause of skin cancer increase is …’ or something like that is more suitable.
Authors’ answer:
We have rephrased this sentence. It now reads, “This development was mainly caused by …”
The reference which is referred to in lines 35-39 is rather old (1995) and the statement is not complete. Not all skin cancers are caused by intermittent exposure. Childhood sun exposure and sunburn are more strongly correlated with melanoma, intermittent exposure is strongly connected with melanoma and BCC, while cumulative exposure is more strongly related to SCC. There are more novel references to point out.
Authors’ answer:
We agree that the reference (the review by Arthey and Clarke) is a little outdated. We added the more recent review paper by Reynolds. We also clarified that intermittent sun exposure is only one of the risk factors contributing to the development of skin cancers by rephrasing this part (see line 36-42).
- Line 40: it sounds as if skin cancer prevention has already become one of the most important health problem issues in European countries, while this should become one of the most important health related issues since many countries do not have established prevention programmes. Further, references are missing.
Authors’ answer:
To our opinion, skin cancer prevention is nowadays one of the most important public health problems; however, we agree with the reviewer that some countries have not recognized this. We rephrased this passage and added further references to clarify the issue, see line 43-46.
- Line 51: intuitively or impossible? We can not see nor feel the proportion of UVI.
Authors’ answer:
We rephrased the sentence. It now reads, “As UV radiation cannot be perceived directly by humans, identifying those outdoor situations in which sun protection is inevitable and finding the appropriate protection level intuitively is hardly possible.”
- It would be worthwhile to read more about the reason why tanned skin as a subject was an important part of the questionnaire.
Authors’ answer:
The questionnaire was designed to assess information on a variety of issues related to sun protection. Attitudes towards tanning and tanned skin were part of the questionnaire as we had observed a strong association between attitudes towards tanning and sun-protective behavior in an earlier study among parents of young children (Gefeller et al., The impact of parental knowledge and tanning attitudes on sun protection practice for young children in Germany. IJERPH 2014;11:4768-4781). To the best of our knowledge, the relationship between UVI-related variables and tanning attitudes has not been addressed in any study before. We incorporated results on this relationship in the manuscript, although we did not observe a clear association as we think that even the non-existence is worth being mentioned given that tanning attitudes have been shown to influence other forms of sun-protective behavior. In the manuscript, we did not stress the motivation of this part of the analysis as we did not find conclusive results. During the revision we decided to refrain from including more information on this issue.
Methods
In general: the method section is well-written, coherently and in-depth. It is very good that the investigators were able to include many kindergartens.
Authors’ answer:
We thank the reviewer for his/her positive evaluation.
What are ‘trained interviewers’? How were they trained? Were they experienced? Were they part of the investigator’s team?
Authors’ answer:
Altogether thirteen different persons were involved as interviewers in the study (the coauthor SM was one of them, the other names are given in acknowledgement section). Most interviewers were medical students or physicians in training, however, two staff members and a student of psychology were also involved. Only a few of them had experience in conducting personal interviews using structured questionnaires. Therefore, training sessions for all interviewers were organized during the study preparations. During the training sessions the questionnaire was presented and discussed in detail. The technique of conducting valid personal interviews – meaning that the interviewer stays absolutely neutral and avoids guiding the interviewed person to a desired answer – was practically rehearsed. Additionally, three selected kindergartens participated in the pilot phase of the study to check feasibility of the study procedures, applicability of the guideline for the structured interviews, and comprehensibility of the questions with directors and staff members of these kindergartens. Information on this issue is given in the manuscript in subsection 2.1 and in the limitations paragraph in the Discussion section.
Results
The study provides important insight into knowledge levels about UVI and attitudes about tanned skin that should be addressed in the future.
It is not clear why the information concerning administrative and financial background of the kindergartens is mentioned, unless the authors would elaborate on it or interpret it further in the manuscript
Authors’ answer:
We felt that reporting these characteristics is a necessary part of the description of our study sample. Although no further analyses using this variable were performed (due to small sample size of several subgroups), we left the descriptive information in the revised version. If the reviewer insists on removing this part, we are prepared to do so in a second revision.
‘… female (96%), only 19 (=4%) male’ (line 88): a typo.
Authors’ answer:
We have corrected the typo.
I would be interested in the percentage of correct answers about specific skin cancer risk factors; which ones stood out?
Authors’ answer:
We thank the reviewer for his/her interest. We have compiled the information in the Table S1, which is now part of the supplementary material available online.
Discussion
In general: the discussion focuses on important findings. The discussion bears some unnecessary repetitive sentences, especially in the second and third paragraph – some parts can therefore be adjusted and/or shortened.
Authors’ answer:
We have rewritten parts of the Discussion section to avoid repetitions. We would, however, like to stress that the second paragraph of the Discussion briefly summarizes the general findings from the systematic reviews on the UVI based on studies in different populations, whereas the third paragraph specifically addresses the setting of kindergartens. Merging these two paragraphs to shorten the Discussion thus doesn’t seem advisable.
‘… The report of the most’ (line 321): this statement is rather logic; improving knowledge does rarely lead to behavior change. The authors could explore this a bit more.
Authors’ answer:
We agree that it is not specific problem of the UVI. When rephrasing parts of the Discussion section we included this aspect (see line 334-339).
Since the difference between self-assessed and actual UVI knowledge is so important, I would like to read more about which misinterpretations or misconceptions were observed among the directors. With that information, a more tailored approach in interventions could be accomplished.
‘disastrous knowledge’ (line 348): again, if the knowledge gaps were so profound and the authors advice a tailored approach, I want to know more about the actual misconceptions.
Authors’ answer:
We share the reviewer’s view that the difference between self-assessed and actual UVI knowledge found in our study is an important - if not the important - finding that has also consequences for the assessment of UVI knowledge in future studies. Future studies have to avoid assessing UVI knowledge by a simple self-assessment of study participants. The personal interviews with the directors were not audio-taped and the interviewers had no instructions to record details of the wrong UVI answers by the directors. The interviewers had to check whether the director mentioned the necessary elements of the UVI definition enabling the classification as a correct answer (see Methods section 2.2). Thus, we have no detailed data on how often the directors made which type of error; only anecdotal evidence from discussions with the interviewers is available. This anecdotal evidence is in line with the results of our qualitative study on UVI-related variables that has been performed among pharmacists (see Diehl K, Görig T, Jansen C, Hruby MC, Pfahlberg AB, Gefeller O. "I've Heard of It, Yes, but I Can't Remember What Exactly It Was"-A Qualitative Study on Awareness, Knowledge, and Use of the UV Index. Int J Environ Res Public Health. 2021 Feb 8;18(4):1615. doi: 10.3390/ijerph18041615.). We incorporated additional explanations on this topic in the revised manuscript, see line 379-387.
Greater parts of the last paragraph of the discussion (before limitations) would actually suit better in the introduction section – where the importance of protection of children at daycare is now quite restricted. Moreover, this part does not elaborate on findings in particular but is more an introductory part for selecting daycare staff as target group.
Authors’ answer:
We agree that this part of the original manuscript could have been placed in the Introduction section, but we consider this a matter of taste and not a must. In the revised version we expanded this paragraph (as a reaction to the remark by another reviewer) now providing some additional reflections on changes of future sun protection policies in kindergartens (see line 401-407). This additional part would be misplaced in the Introduction and can only be part of the Discussion. Therefore, we refrained from rearranging the manuscript.
I am missing reflection on a highly important finding; the high percentage of directors that saw tanned skin as beautiful (n=292) and as healthy (n=86). The attitude about tanned skin is now absent in the discussion section, while it is 1) a very important finding and can function as a barrier for actual sun protection behavior, and 2) a topic which has gained a lot of attention in earlier skin cancer prevention research. The authors should implement a section which discusses this imperative topic.
Authors’ answer:
The results regarding the attitudes towards tanned skin among directors of kindergartens were intentionally left out in the Discussion section. The manuscript focuses on the UVI and the relationship of UVI-related variables to other characteristics. We did not find a consistent pattern of results linking the attitudes towards tanned skin to the UVI-related variables in our study. Such a relationship has not been observed in earlier studies (as it had not been examined) which means that our results are not in conflict with earlier findings. To our opinion, the non-existence of a relationship that has not been found by others before is not a relevant part of a Discussion section. In order to give more emphasis to our data on attitudes towards tanned skin among directors of kindergartens we expanded the start of section 3.7 to provide more descriptive details, but we refrained from incorporating this issue in the Discussion section.